# Nujiangexanthone A Inhibits Cervical Cancer Cell Proliferation by Promoting Mitophagy

**DOI:** 10.3390/molecules26102858

**Published:** 2021-05-12

**Authors:** Jiling Feng, Anahitasadat Mansouripour, Zhichao Xi, Li Zhang, Gang Xu, Hua Zhou, Hongxi Xu

**Affiliations:** 1Institute of Cardiovascular Disease of Integrated Traditional Chinese and Western Medicine, Shuguang Hospital affiliated to Shanghai University of Traditional Chinese Medicine, No. 528, Zhangheng Road, Shanghai 201203, China; fjlfreda@shutcm.edu.cn; 2School of Pharmacy, Shanghai University of Traditional Chinese Medicine, No. 1200, Cailun Road, Shanghai 201203, China; Anahita.mp2009@gmail.com (A.M.); xizhichao@shutcm.edu.cn (Z.X.); 0000002736@shutcm.edu.cn (L.Z.); 3State Key Laboratory of Phytochemistry and Plant Resources in West China and Yunnan Key Laboratory of Natural Medicinal Chemistry, Kunming Institute of Botany, Chinese Academy of Sciences, Kunming 650201, China; xugang008@mail.kib.ac.cn

**Keywords:** nujiangexanthone A, mitophagy, ATG7, starvation

## Abstract

Nujiangexanthone A (NJXA), a bioactive component isolated from the leaves of *Garcinia nujiangensis*, has been reported to exhibit anti-inflammatory, antioxidant, and antitumor effects. Our previous work has shown that NJXA induced G_0/1_ arrest and apoptosis, thus suppressing cervical cancer cell growth. The present study provides new evidence that NJXA can induce cell death in HeLa cells by promoting mitophagy. We first identified that NJXA triggered GFP-LC3 and YFP-Parkin puncta accumulation, which are biomarkers of mitophagy. Moreover, NJXA degraded the mitochondrial membrane proteins Tom20 and Tim23 and mitochondrial fusion proteins MFN1 and MFN2, downregulated Parkin, and stabilized PINK1. Additionally, we revealed that NJXA induced lysosome degradation and colocalization of mitochondria and autophagosomes, which was attenuated by knocking down ATG7, the key regulator of mitophagy. Furthermore, since mitophagy is induced under starvation conditions, we detected the cytotoxic effect of NJXA in nutrient-deprived HeLa cells and observed better cytotoxicity. Taken together, our work contributes to the further clarification of the mechanism by which NJXA inhibits cervical cancer cell proliferation and provides evidence that NJXA has the potential to develop anticancer drugs.

## 1. Introduction

Cervical cancer remains the fourth leading cause of cancer in women worldwide and accounted for 569,847 new cases and 311,365 deaths in 2018 [1]. Although techniques for diagnosis and treatment have developed significantly during the past decades, the incidence of cervical cancer has continued to increase in China, most probably due to the lack of organized screening and the low penetration rate of human papillomavirus (HPV) vaccines. Current treatments for cervical cancer mainly focus on hysterectomy, pelvic lymphadenectomy, radiotherapy, and chemotherapy [2]. Cisplatin-containing concurrent chemotherapy is superior in women with advanced cervical cancer. However, cisplatin does not perform to its highest potential because of drug resistance and side effects, such as nephrotoxicity, ototoxicity, hepatotoxicity, gastrointestinal toxicity, and neurotoxicity [3]. Therefore, new therapeutic strategies and effective drugs without severe side effects are currently being explored. Among them, the most popular research has focused on anticancer screening of natural products and the development of novel drug delivery systems.

Mitochondria play a central role in cellular energy metabolism, producing approximately 95% of adenosine triphosphate (ATP). Mitochondrial quality is associated with the synthesis of new mitochondria, which are regulated by the conserved processes of fusion, fission, and mitophagy [4]. Mitophagy is the process of eliminating cellular waste and injured mitochondria, which is essential for maintaining mitochondrial stability. In 2005, Lemasters first identified “mitophagy” as a separate event to autophagy that specifically targets mitochondria when cells are in hostile microenvironments, such as high reactive oxygen species (ROS) levels, nutrient deprivation, and cellular senescence [5]. Damaged mitochondria are wrapped into autophagosomes and fused with lysosomes, thus degrading injured organs and maintaining the stability of the inner cellular environment. Mitophagy plays an important role in mitochondrial dynamics and is vital for the quantity and quality of mitochondria, as well as cell differentiation and certain diseases.

Dysfunctional mitochondria are commonly observed in cancer cells, and most mitophagy receptors and regulators are involved in cancer [5]. Moreover, homogeneous mutations in the mitochondrial genome have been found in primary tumors and are closely related to their growth and metastasis [6]. Proteins responsible for mitophagy, such as PINK1, Parkin, BNIP3, NIX, and p62, have been implicated in various cancers [4]. The PINK1–Parkin pathway is not only the first but also the most well-studied signaling pathway related to mitophagy [7]. When mitochondria are damaged or depolarized, PINK1 is stabilized on the outer mitochondrial membrane and recruits and activates its ubiquitin protein ligase Parkin. Parkin is transported from the cytoplasm to the mitochondria, and then ubiquitinates the proteins on the outer mitochondrial membrane, such as TOM20, MFN1, and MFN2, thereby recruiting P62 and optineurin (OPTN). These proteins then bind to autophagosome proteins (mostly LC3), directing phagophores on the autophagosome to surround the damaged mitochondria [8]. Finally, the enclosed mitochondria are degraded and eliminated.

Nujiangexanthone A (NJXA) is a bioactive compound isolated from *Garcinia nujiangensis*, which exhibited anticancer [9,10] and anti-inflammatory [11] activities in our previous work. Here, we screened novel mitophagy inducers from several xanthones of *Garcinia* species using human cervical carcinoma HeLa cells stably expressing YFP-Parkin. This work contributes to the mechanistic study of NJXA in inducing cancer cell death and evaluates the potential of NJXA to be developed into an anticancer drug.

## 2. Results

### 2.1. NJXA Induces Puncta Formation of Parkin in HeLa Cells

To identify small molecules that can regulate mitophagy, we previously established a HeLa cell line stably transfected with YFP-Parkin. Parkin is an important E3 ubiquitin ligase that localizes throughout the nucleus and cytosol under normal conditions and translocates to damaged mitochondria to trigger mitophagy under stress conditions [7]. We treated YFP-Parkin HeLa cells with several xanthones from Garcinia species and found that NJXA significantly induced the formation of YFP-Parkin puncta (data not shown). Next, we performed a time-dependent observation using a fluorescence microscope and found that NJXA induced puncta accumulation in a time-dependent manner (Figure 1A). To explore the possibility of NJXA-induced Parkin-dependent mitophagy, we detected mitophagy-related proteins using immunoblotting analysis. NJXA significantly stabilized PINK1 and promoted Parkin degradation. The mitochondrial outer membrane protein Tom20 and inner membrane protein Tim23 as well as the fusion proteins MFN1 and MFN2 were also degraded after NJXA treatment. These results suggest that NJXA induces mitophagy in HeLa cells, probably in a Parkin-dependent way.

### 2.2. NJXA Depolarizes Mitochondria in HeLa Cells

The reduction in mitochondrial membrane potential (MMP) in damaged mitochondria leads to the accumulation and activation of PINK1 on the outer mitochondrial membrane and activates mitophagy [12]. Therefore, we detected MMP in NJXA-treated HeLa cells. As shown in Figure 2A,B, NJXA treatment for 2 h resulted in the loss of MMP. When PINK1-activated Parkin is recruited to mitochondria, it modulates the process of mitophagy by causing a decrease in mitochondrial mass, finally resulting in the elimination of damaged mitochondria [13]. MitoTracker, a fluorescent dye directly accumulated on active mitochondria, was used to detect the mass of mitochondria. As shown in Figure 2C, NJXA markedly reduced the red fluorescence intensity of MitoTracker Red, indicating decreased mitochondrial mass. Although CCCP (carbonyl cyanide 3-chlorophenylhydrazone) is a widely accepted mitophagy inducer, the effect of CCCP on decreasing mitochondrial mass of HeLa cells was not significant as expected. On the contrary, the effect of NJXA on decreasing MitoTracker fluorescence was far more significant, indicating that NJXA was a better mitophagy inducer than CCCP from this aspect. Taken together, NJXA depolarizes mitochondria and initiates mitophagy in cervical cancer HeLa cells.

### 2.3. NJXA Promotes p62 Recruitment to Mitochondria

Upon mitochondrial depolarization, PINK1 stabilizes on the outer mitochondrial membrane and recruits its RBR-type E3 ubiquitin ligase Parkin. Parkin is phosphorylated and then ubiquitinates proteins, such as the mitochondrial fusion proteins MFN1, MFN2, and voltage-dependent anion channels (VDACs) [12,14]. MFN2 localizes on the outer membrane of mitochondria and can be used to indicate the mitochondrial position. To address whether NJXA promotes Parkin recruitment to mitochondria, we treated YFP-Parkin HeLa cells with NJXA. As demonstrated in Figure 3A, NJXA significantly induced colocalization of Parkin (green) and MFN2 (red), suggesting that it promoted Parkin transfer to mitochondria. The autophagic protein p62 has been reported to be a missing link between ubiquitylation and mitophagy, whereby p62 binds to LC3 on the phagophore and modulates mitophagy [15]. We next investigated whether NJXA promoted the recruitment of p62 and LC3 to the mitochondria. Treatment with NJXA significantly enhanced the colocalization of Parkin with p62 and LC3, suggesting that the autophagy receptor p62 is recruited to damaged mitochondria to drive mitophagy (Figure 3B,C). Taken together, these data suggest that NJXA promotes Parkin-dependent mitophagy following the recruitment of the autophagy receptor p62.

### 2.4. NJXA Induces LC3 Puncta Formation and Promotes Mitochondria–Lysosome Fusion

As autophagosome accumulation can be detected by observing LC3 puncta formation, we observed LC3 conversion in NJXA-treated GFP-LC3 HeLa cells. Fluorescence microscopy images revealed that GFP puncta accumulated after NJXA treatment in a time-dependent manner (Figure 4A). LC3-II is specifically associated with autophagosome membranes, and the transformation from LC3-I to II is correlated with the extent of autophagosome formation [16]. As shown in Figure 4B,C, NJXA treatment increased the amount of LC3-II/I in a time-dependent manner, which is consistent with the fluorescence analysis findings. As we already identified that NJXA induced colocalization of Parkin and LC3, these results confirm that NJXA induced mitochondria–autophagosome fusion.

### 2.5. NJXA Promotes Mitochondria–Lysosome Fusion without Activating Lysosome Functions

Activated Parkin and autophagic proteins can promote mitochondrial movement along microtubules, as well as autophagosome–lysosome fusion [6]. To address whether NJXA affects mitochondria and lysosome fusion, we examined the colocalization of Parkin and LAMP1, a lysosomal membrane protein. As shown in Figure 3D, the yellow puncta indicated colocalization of Parkin and lysosomes.

Next, we observed lysosome biogenesis by detecting lysosomal proteins and enzymes. Cathepsins are ubiquitously expressed lysosomal aspartyl proteases involved in protein degradation, revealing the activity of lysosomal biogenesis [17]. We detected cathepsin D and membrane protein LAMP1 in HeLa cells by immunoblotting and found that both were degraded after NJXA treatment (Figure 4D). Results of the enzymatic activity assay also illustrated that cathepsin L activity was significantly decreased upon NJXA treatment, but increased by positive control drugs (Figure 4E). PP242 (Torkinib) and torin-1 were used as positive drugs as they are known activators of lysosomal activity [18]. LysoTracker Red staining revealed that NJXA significantly blocked the fluorescence of LysoTracker Red (Figure 4F), while PP242 strengthened it, indicating that NJXA downregulated lysosomal activity. Together, these results suggest that NJXA triggers mitochondrial and lysosomal fusion, subsequently inducing the removal of damaged mitochondria instead of activating lysosomal function.

### 2.6. NJXA-Induced Mitophagy and Cell Death Are Reversed by ATG7 Knockout

ATG7 regulates autophagosome elongation and plays an important role in mitophagy by manipulating autophagic flux [19]. We knocked down ATG7 using lentiviral transfection in HeLa, YFP-Parkin HeLa, and GFP-LC3 HeLa cell lines. As shown in Figure 5A, NJXA-induced cell proliferation was alleviated in ATG7 knockdown HeLa cells after a 72 h treatment. As the effect of NJXA on cell proliferation was blocked, we sought to determine whether other mitophagy-related procedures were affected. The results of immunoblotting showed that the expression of mitophagy-related proteins changed after ATG7 knockdown in HeLa cells, and NJXA-induced changes in these protein levels were partially abrogated (Figure 5B). Flow cytometry with JC-1 staining showed that NJXA could no longer decrease the MMP after ATG7 knockdown (Figure 5C). Results of immunofluorescence confirmed that NJXA-induced colocalization of mitochondria and lysosomes was abolished. In contrast to HeLa-ctr (cells transfected with empty vector) cells, YFP-Parkin puncta were hardly detected after ATG7 knockdown. In HeLa-ATG7 cells, treatment with NJXA had no effect on the colocalization of Parkin with MFN2, p62, and LAMP1, illustrating that NJXA could not promote mitochondrial, autophagosome, and lysosome fusion. Taken together, NJXA-promoted proliferation inhibition and mitophagy induction were both attenuated by ATG7 knockdown, illustrating that mitophagy is at least partially responsible for the anticancer effect of NJXA on cervical cancer cells.

### 2.7. NJXA Eliminates the Tolerance of Cancer Cells to Nutrient Starvation.

It has been reported that nutrient deprivation induces mitophagy [20]. To further confirm that NJXA induces cervical cancer cell death by inducing mitophagy, we treated nutrient-deprived HeLa cells with NJXA. The results of the CCK-8 (Cell Counting Kit-8) assay suggest that NJXA was more sensitive to HeLa cells under starvation conditions (Figure 6A). The difference between the IC_50_ (50% inhibition concentration) of NJXA in complete medium and nutrient-deprived conditions is shown in Figure 6B. Following a 72 h treatment, for example, NJXA (3.75 μM) had nearly no effect on complete-medium-cultured HeLa cells but inhibited more than 90% of cell proliferation under starvation conditions. To further detect the effect of NJXA on mitophagy induction under starvation conditions, immunofluorescence was performed. As shown in Figure 6C, we observed colocalization of Parkin with MFN2, p62, and LAMP1. The yellow section of colocalization analysis showed a nutrient deprivation-enhanced effect of NJXA. These results suggest that NJXA inhibited cervical cancer cell proliferation under nutrient starvation even more effectively than sufficient nutrient supply, and the mechanism may be attributed to enhanced mitophagy under starvation conditions.

## 3. Discussion

Mitochondrial dysfunction is a distinctive feature of tumor cells. When cancer cells are under stress, such as nutrient deficiency, they use raw materials in host cells to provide energy and adjust cell metabolism through autophagy [21]. Therapies that target tumor cell mitochondria can effectively release the coupling between tumor cells and the “parasitic host,” cutting off the energy source of tumor cells. Mitophagy is correlated with various cancer cell characteristics, including the promotion of ROS production, disrupting the redox balance [22], accelerating the mitochondrial unfolded protein response (UPRmt), increasing the production of unfolded proteins and toxic metabolites [23], increasing mitochondrial membrane permeability, promoting cytochrome C release into the cytoplasm to induce cell apoptosis [24], adjusting intracellular energy metabolism to cause drug resistance [25], and inducing mitochondrial DNA damage [26]. Since mitophagy is generally induced in cancer cells, therapeutic strategies targeting mitophagy possess better selectivity and efficiency, which can be a direction for anticancer drug screening.

To date, there have been some antitumor studies on natural products based on mitophagy, such as artemisinin, berberine, icariin, stephentanin, and liensinine. Among them, artemisinin regulates mitochondrial membrane permeability, promotes cytochrome C release, induces overproduction of ROS by mediating the PINK1–Parkin pathway, and shows selective anticancer properties in cervical, colon, and breast cancer cells [27]; by binding to the GPR30 receptor and regulating the mitochondrial fission process, stephentanin hydrochloride induces PINK1–Parkin pathway-dependent mitochondrial autophagy and inhibits the proliferation of liver cancer cells [28]. Icariin degrades mitochondrial DNA, reduces the protein expression of the PINK1–Parkin pathway, induces mitochondrial fragmentation, and acts synergistically with adriamycin to induce mitophagy and apoptosis to treat therapy-resistant hepatocellular carcinoma [29]. Berberine induces mitophagy-dependent necrosis and inhibits cancer cell proliferation by inhibiting the expression of PCYT1A in diffuse large B-cell lymphomas [30]. Liensinine affects the dephosphorylation and mitochondrial metastasis of DNM1L, which inhibits the recruitment of RAB7A to lysosomes, thereby blocking autophagosome–lysosome fusion and enhancing doxorubicin-mediated apoptosis, ultimately reducing the activity of breast cancer cells [31]. The above results show that regulating mitochondrial autophagy by regulating the PINK1–Parkin pathway is one of the ways to break through the bottleneck of cervical cancer treatment. Natural products have the potential for the development of antitumor drugs owing to their advantages of alleviating side effects and multitarget therapy.

Previous studies have demonstrated the various antitumor and anti-inflammatory bioactivities of NJXA. For example, NJXA inhibited the expression of heterogeneous ribonucleoprotein K (hnRNP K) by promoting the degradation of the ubiquitin–proteasome-dependent pathway, finally resulting in cell cycle arrest [9]. NJXA could activate the JNK pathway to promote ROS production, thus inducing caspase-3-dependent apoptosis [10]. Moreover, NJXA suppressed IgE/Ag-induced mast cell activation by inhibiting Src kinase activity and Syk-dependent pathways, and substantially inhibited OVA-induced cellular infiltration [11]. In the present study, we found that NJXA effectively triggered GFP-LC3 and YFP-Parkin puncta accumulation and degraded proteins related to the PINK1–Parkin pathway. NJXA defuncted mitochondria and lysosomes and induced colocalization of mitochondria, autophagosomes, and lysosomes. The effect of NJXA on inducing mitophagy and inhibiting cancer cell proliferation was abolished by knocking down the mitophagy regulator ATG7 and enhanced by the mitophagy-induced environment induced by nutrient starvation. Our results suggest that NJXA inhibits cervical cancer proliferation by inducing mitophagy.

Notably, our results also show that by blocking mitophagy by knocking down the autophagic protein ATG7, the effect of NJXA on inhibiting cell proliferation could not be completely eliminated, suggesting that NJXA inhibited cervical cancer cell proliferation in a multiple-target manner, and that the induction of mitophagy is only one of the mechanisms by which NJXA works.

The structure and function of mitochondria are maintained through a balance between biogenesis and mitophagy, mitochondrial fusion, and fission. Reduced mitochondrial fusion causes greater autonomy for individual organelles in the mitochondrial population, a state that increases heterogeneity among organelles and results in dysfunction [32]. In mammals, mitochondrial fusion is controlled by MFN1 and MFN2. We showed that knocking down MFN2 alone did not have an obvious effect on cell proliferation or on the inhibitory effect of NJXA (data not shown). It has been reported that cells null for only MFN1 and MFN2 contain low but readily measured rates of fusion; MFN-null cells have completely fragmented mitochondria and show no detectable fusion [32]. This may be attributed to the overlapping expression patterns of MFN1 and MFN2, which share 80% of the sequence [32,33]. To further study how mitochondrial dynamics play a role in the anticancer effect of NJXA, we will consider the knockout of both mitofusion proteins in future work.

Cancer cells have an inherent ability to tolerate hostile microenvironments, such as low nutrient supplies and hypoxia, by modulating the energy metabolism pathway [34]. Therefore, finding new drugs that are effective under conditions of nutrient deprivation is worthwhile. Autophagic degradation products can be reused for protein synthesis and energy production [8]. Mitophagy is a selective type of autophagy targeting mitochondria, which serves as the main source of energy for cancer cells under starvation. Furthermore, inducing mitophagy in cancer cells may lead to excessive self-engulfment of mitochondria, restricting the energy supply to cancer cells, and ultimately result in cancer cell death. According to our results, further induction of mitophagy to break down hemostasis can be used to treat cervical cancer cells under conditions of nutrient deprivation. On the other hand, cells cultured in complete medium did not respond to NJXA treatment at low dosage, while at the same dosage, cell proliferation under nutrient starvation was significantly inhibited. These results imply that NJXA selectively targets these cells under starvation conditions and might not be toxic to cervical cancer cells under normal nutrient environments at low concentrations. Tolerance to nutrient starvation might be a part of the biological response to insufficient blood supply, and targeting mitophagy to eliminate cancer cells under conditions of nutrient deprivation may be a novel approach in anticancer drug development.

## 4. Materials and Methods

### 4.1. Compound

Nujiangexanthone A (with a purity greater than 98%) was extracted and purified from the leaves of *Garcinia nujiangensis* in our laboratory [35].

### 4.2. Cell Lines and Culture Methods

HeLa, GFP-LC3 HeLa cells, and YFP-Parkin HeLa cells were maintained in Dulbecco’s modified Eagle’s medium (DMEM, Meilunbio, Dalian, China) containing 10% fetal bovine serum (Biological Industries, Haemek, Israel) and 1% penicillin/streptomycin (Genom, Zhejiang, China) at 37 °C in a humidified incubator with 95% air and 5% CO_2_. For nutrient starvation, HeLa cells cultured in DMEM were washed three times with phosphate buffer saline (PBS) and then cultured in DMEM without fetal bovine serum for the indicated time points.

### 4.3. Immunoblotting

Immunoblotting was performed as previously described [36]. Cells were lysed with ice-cold RIPA lysis buffer in the presence of a protease inhibitor cocktail. Protein quantification, electrophoresis, and transfer were performed, followed by membrane blocking with 5% nonfat milk. The human-specific antibodies used included GAPDH (ab128915) from Abcam; Tom20 (sc-11415), Parkin (sc-133167), and Tim23 (sc-514463) from Santa Cruz; p62 (23214), PINK1 (6946), and LC3 (3868) from Cell Signaling Technology (Boston, MA, USA); and MFN1 (66776-1-Ig) and MFN2 (67487-1-Ig) from Proteintech (Chicago, IL, USA). Then they were incubated with secondary antibodies (KPL) for 1 h at room temperature. Protein bands were visualized using ECL blotting detection reagents (Life-iLab, Shanghai, China).

### 4.4. YFP Translocation and Colocalization Analyses

YFP-Parkin HeLa cells (1 × 10^5^) were grown on coverslips and treated with 20 μM NJXA for the indicated times. The cells were then fixed in 4% paraformaldehyde and permeabilized with 0.3% Triton-X. Cells were blocked with 0.5% BSA for 0.5 h at room temperature. After overnight incubation with primary antibodies (1:500) at 4 °C, cells were washed with PBS and incubated with a Cy3-conjugated secondary antibody (1:500, Beyotime, Shanghai, China) at room temperature for 3 h. All fluorescence images were acquired using a fluorescence microscope (Olympus, Tokyo, Japan).

### 4.5. Live Cell Imaging

GFP-LC3 HeLa and YFP-Parkin HeLa cells (1 × 10^5^) were cultured in glass-bottom 24-well plates. After treatment with NJXA for the indicated times, cells were washed with PBS three times and examined under a fluorescence microscope (Olympus, Tokyo, Japan).

### 4.6. LysoTracker Staining

LysoTracker staining was performed as described previously [37]. HeLa cells (2 × 10^5^) were first treated with NJXA and then stained with 50 nM LysoTracker Red (Invitrogen, MA, USA) in prewarmed medium for 20 min at 37 °C. Labeled cells were photographed under a fluorescence microscope, harvested, and measured by flow cytometry (BD Biosciences, San Jose, CA, USA).

### 4.7. Lentiviral Transfection

ATG7 and empty vector plasmids were purchased from GeneChem. Plasmids were transfected into Hek-293T cells using a 2nd Generation Packaging System Mix (GeneCopoeia, Rockville, MD, USA) according to the manufacturer’s protocol. Viruses were collected and stored at −80 °C. HeLa, GFP-LC3 HeLa, and YFP-Parkin HeLa cells (5 × 10^4^) were seeded in 24-well plates and infected with ATG7 and control lentivirus for 72 h according to the manufacturer’s protocol.

### 4.8. Mitochondrial Membrane Potential Detection

The mitochondrial membrane potentials were assessed using a JC-1 staining assay kit (Beyotime, Shanghai, China). HeLa cells (2 × 10^5^) were first treated with NJXA and then stained with JC-1 at 37 °C for 20 min. Cells were then washed three times with ice-cold PBS and collected for flow cytometry analysis.

### 4.9. MitoTracker Staining

After treatment with NJXA for 0–4 h, HeLa cells (2 × 10^5^) were incubated with 100 nM MitoTracker Red (Thermo Fisher, Waltham, MA, USA) in the dark for 30 min at 37 °C. Cells were then washed with precooled PBS. The fluorescence of the cells was analyzed by flow cytometry.

### 4.10. CCK-8 Assay and IC_50_ Calculation

Cell proliferation assays using CCK-8 staining were performed as described previously [38]. Briefly, HeLa cells (5 × 10^3^) were seeded in a 96-well plate and treated with NJXA for 0–72 h. After treatment, 10 μL of CCK-8 solution (TargetMol, Boston, MA, USA) was added and incubated with the cells for 1 h.

After careful mixing, the absorbance was measured at 570 nm, and cell viability was normalized by expressing each absorbance value as a percentage of the control value. The IC_50_ values were determined using SPSS software 21.0, (SPSS, Inc., Chicago, IL, USA).

### 4.11. Cathepsin L Enzymatic Activity

The enzymatic activity of cathepsin L was assessed using the Cathepsin L Assay Kit (Abcam). HeLa cells (2 × 10^5^) treated with NJXA were incubated with a Magic Red cathepsin L substrate for 15 min, before cell fluorescence of the cells was analyzed by flow cytometry.

### 4.12. Statistical Analysis

All results are expressed as the mean ± SD of at least three independent experiments. Statistical analysis was performed using SPSS version 21.0. ANOVA was used with Fisher’s LSD multiple comparison test for multiple groups. Values of * *p* < 0.05, ** *p* < 0.01, and *** *p* < 0.001 were considered statistically significant.

## 5. Conclusions

In conclusion, our findings demonstrate that NJXA partially inhibited cervical cancer cell proliferation by inducing mitophagy. The effect of NJXA on the inhibition of cell proliferation and induction of mitophagy was enhanced under nutrient deprivation, suggesting that NJXA has the potential to combat cancer cells that are tolerant to caloric restriction.

## Figures and Tables

**Figure 1 molecules-26-02858-f001:**
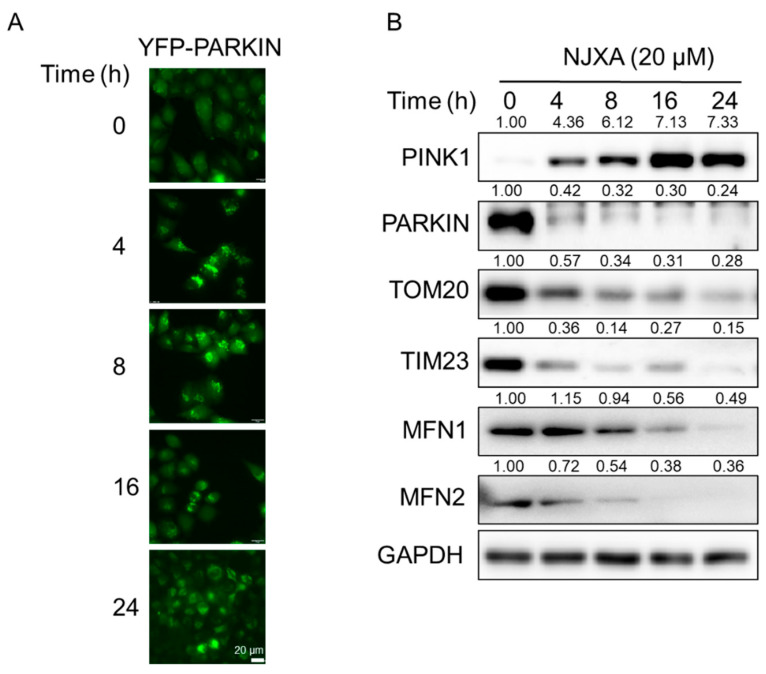
NJXA promotes Parkin-dependent mitophagy in HeLa cells. (**A**) YFP-Parkin HeLa cells were treated with NJXA (20 μM) for 0–24 h, and the distribution of YFP-Parkin was examined by immunofluorescence microscopy. Scale bar = 20 μM. (**B**) HeLa cells treated with NJXA (20 μM) for 0–24 h were analyzed by immunoblotting for PINK1, PARKIN, TOM20, TIM23, MFN1, and MFN2. GAPDH was served as loading control. The values above the bands indicate the relative 669 band intensities. Data are presented as the means ± SD of three independent experiments.

**Figure 2 molecules-26-02858-f002:**
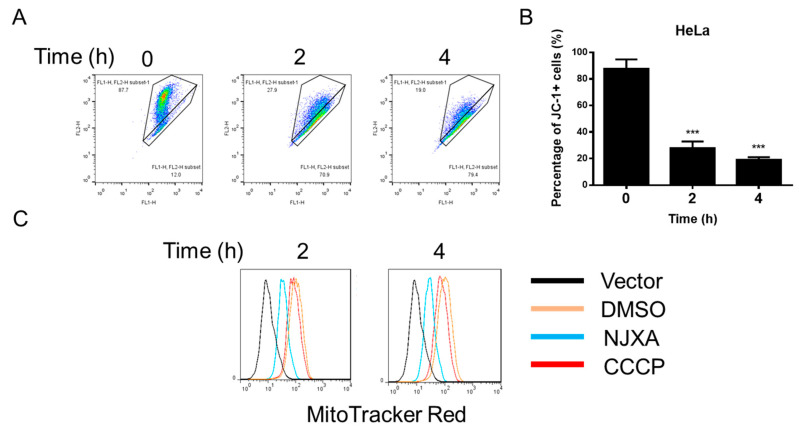
NJXA degrades mitochondria in HeLa cells. (**A**) HeLa cells treated with NJXA (20 μM) for 0, 2, and 4 h were collected for mitochondrial membrane potential detection. (**B**) Quantification of the proportion of JC-1 positive cells of (**A**). *** *p* < 0.001. (**C**) HeLa cells were treated with NJXA (20 μM) or CCCP (20 μM) for 0, 2, and 4 h; labeled with MitoTracker Red (100 nM); and analyzed by immunofluorescence and flow cytometry.

**Figure 3 molecules-26-02858-f003:**
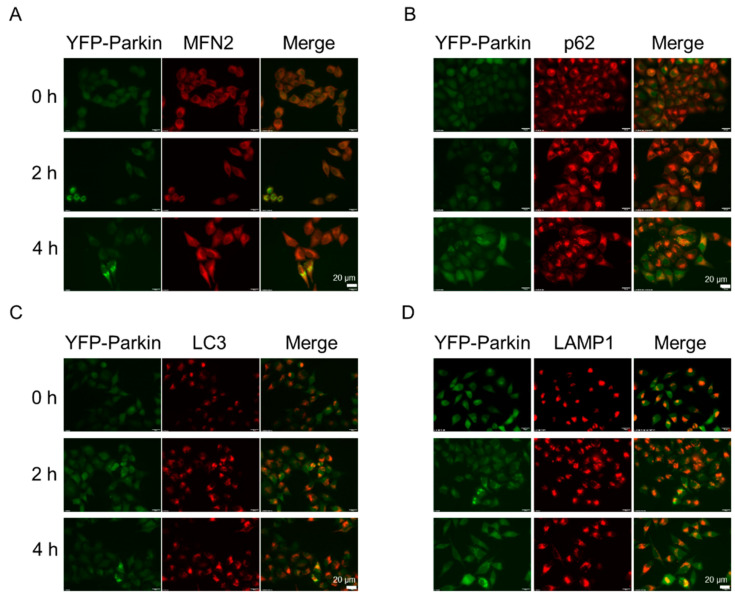
NJXA promotes mitochondria–autophagosome fusion. YFP-Parkin HeLa cells were treated with NJXA (20 μM) for 4 h, fixed, and stained with (**A**) MFN2, (**B**) p62, (**C**) LC3, and (**D**) LAMP1 antibodies. The images were acquired using a fluorescence microscope. Scale bar = 20 μM.

**Figure 4 molecules-26-02858-f004:**
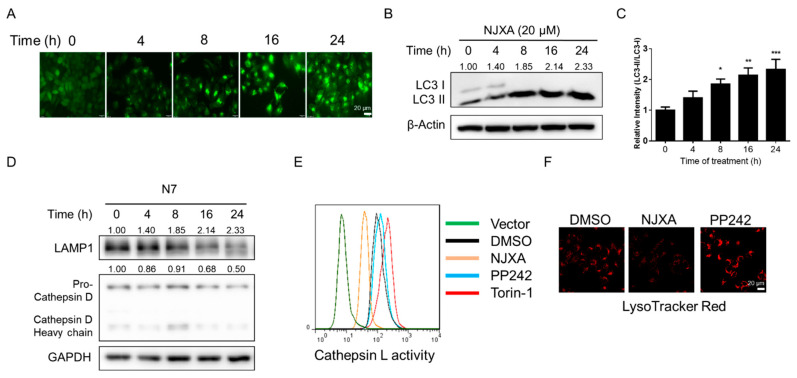
NJXA promotes mitochondria–lysosome fusion without activating lysosomal function. (**A**) GFP-LC3 HeLa cells were treated with NJXA (20 μM) for 0–24 h. The distribution of GFP-LC3 was examined by immunofluorescence microscopy. Scale bar = 20 μm. (**B**) HeLa cells treated with NJXA (20 μM) for 0–24 h were analyzed by immunoblotting for LC3, and β-actin served as the loading control. (**C**) ImageJ densitometric analysis of the LC3-II/LC3-I ratio from the immunoblots is shown (mean ± SD of three independent experiments). * *p* < 0.05, ** *p* < 0.01, and *** *p* < 0.001. (**D**) HeLa cells were treated with NJXA (20 μM) for 0–24 h, collected, and analyzed by immunoblotting for LAMP1 and cathepsin D. GAPDH served as the loading control. The values above the bands indicate the relative 669 band intensities. Data are presented as the means ± SEM of three independent experiments. HeLa cells were treated with NJXA (20 μM), PP242 (1 μM), or torin-1 (1 μM) for 4 h. (**E**) Cells were labeled with Magic Red Cathepsin L reagent for 30 min and assessed by flow cytometry. (**F**) Alternatively, cells were labeled with LysoTracker Red (50 nM) and analyzed by fluorescence microscopy.

**Figure 5 molecules-26-02858-f005:**
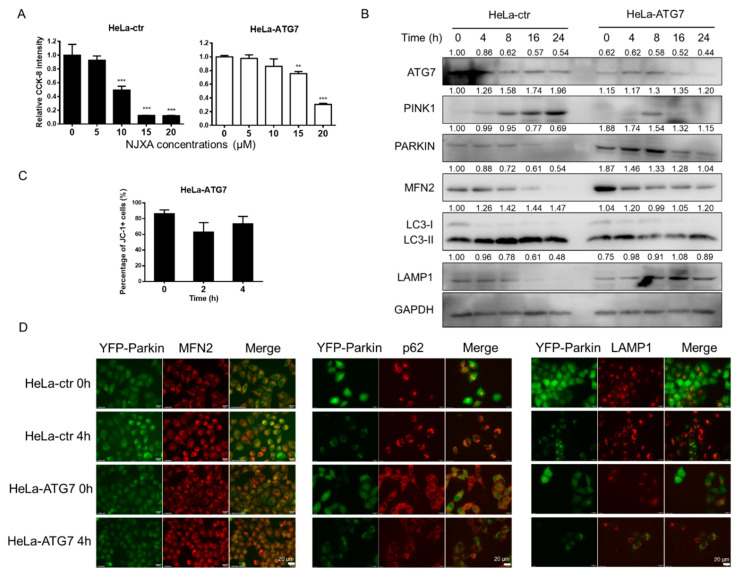
Effect of NJXA on HeLa cells is partially reversed by mitophagy inhibition. (**A**) HeLa-ctr and HeLa-ATG7 cells were treated with NJXA at the indicated concentrations (0–20 μM) for 72 h and evaluated by CCK-8 assay. Data is shown as mean ± SD of three independent experiments, ** *p* < 0.01, and *** *p* < 0.001. (**B**) HeLa-ctr and HeLa-ATG7 cells were treated with NJXA (20 μM) for 0–24 h and analyzed by immunoblotting for ATG7, PINK1, PARKIN, MFN2, LC3, and LAMP1. GAPDH served as the loading control. The values above the bands indicate the relative 669 band intensities. Data are presented as the means ± SEM of three independent experiments. (**C**) HeLa-ATG7 cells treated with NJXA (20 μM) for 0, 2, and 4 h were collected for mitochondrial membrane potential analysis. (**D**) YFP-Parkin HeLa cells transfected with ATG7 or control plasmid cells were treated with NJXA (20 μM) for 4 h, fixed, and stained with MFN2, p62, and LAMP1 antibodies. The images were acquired using a fluorescence microscope. Scale bar = 20 μM.

**Figure 6 molecules-26-02858-f006:**
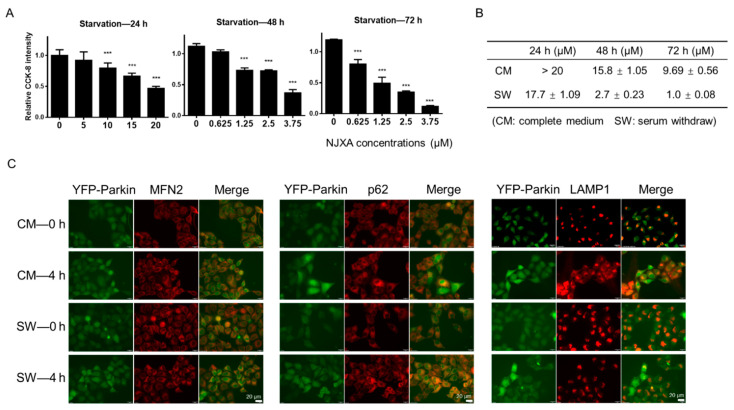
Nutrition deprivation-induced mitophagy enhances sensitivity to NJXA in HeLa cells. (**A**) HeLa cells were treated with NJXA at 0–20 μM for 24 h and 0–3.75 μM for 48 and 72 h, respectively, in DMEM without fetal bovine serum. Cells were analyzed by CCK-8 assay. Data is shown as mean ± SD of three independent experiments, *** *p* < 0.001. (**B**) The IC_50_ values of NJXA were calculated according to the results of the CCK-8 assay. (**C**) YFP-Parkin HeLa cells were treated with NJXA (5 μM) for 4 h in DMEM with or without fetal bovine serum. Cells were fixed and stained with MFN2, p62, and LAMP1 antibodies. The images were acquired using a fluorescence microscope. Scale bar = 20 μm.

## Data Availability

Not applicable.

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
