# Peer review of "Nujiangexanthone A Inhibits Cervical Cancer Cell Proliferation by Promoting Mitophagy"

_molecules, 2021, doi:10.3390/molecules26102858_

Round 1

Reviewer 1 Report

Opinion

on the manuscript by Ji-ling Feng et al., entitled “Nujiangexanthone A inhibits cervical cancer cell proliferation by promoting mitophagy”

Manuscript ID: molecules-1179931

The manuscript of Ji-ling Feng et al. deals with the mechanism of anticancer natural product nujiangexanthone A (NJXA) on Hela cells. The paper is generally well-written, the presented in vitro results are original useful for colleagues working on a related field. Authors utilized a wide array of sophisticated cell-based methods which are generally accepted in similar mechanistic studies. There are, however, some weaknesses for correction or improvement in the current version of the manuscript. These are the followings:

Major points:

  1. A substantial part of the presented results is not evaluated statistically (e.g. data in Figs 1B, 2C, 4D). No relevant conclusion can be stated without statistical evaluation even in the case of substantial differences.
  2. While most treatment-related changes presented in Fig 1B are time-dependent the expression of MFN1 seems to be biphasic. It is substantially decreased after 4-8 h and returned by 24 h. A comment or explanation is expected. Fig 1B is mentioned in the text.
  3. Fig 4B is supposed to be a representative set of data presented in Fig 4C. But according to 4C the LC3-II/LC3-I ratio is approximately 1 for the 0 point which is not supported by 4B.
  4. Many of the methodological descriptions are not sufficient for reproduction of the experiments. E.g. the cell number used for the assay is not given.
  5. Lines 384-386: “A Student's two-tailed t-test was used to compare two different groups …” but there is no presented result in which 2 conditions are compared.

Minor points:

  1. Line 94: “N7 treatment”. What is N7? It is probably a mistake.
  2. The quality of Fig 4E is quite poor. An improved version is expected with higher resolution.
  3. Line 166: definition of OC is missing. Probably it refers to the tested compound.
  4. Some of the cited papers are inefficiently given in the References. In some cases, the journal is missing (e.g. refs 13, 19, 20, 27). A complete and careful checking is suggested.

Based on all these objections the rejection of the current version of the paper is suggested. Reconsideration seems rationale after the correction of the mentioned points.

Author Response

Reviewer1

  1. Q: A substantial part of the presented results is not evaluated statistically (e.g. data in Figs 1B, 2C, 4D). No relevant conclusion can be stated without statistical evaluation even in the case of substantial differences.

A: We agree with your assessment. All statistic evaluation of WB results has already been added on the membranes.

  1. Q: While most treatment-related changes presented in Fig 1B are time-dependent the expression of MFN1 seems to be biphasic. It is substantially decreased after 4-8 h and returned by 24 h. A comment or explanation is expected. Fig 1B is mentioned in the text.

A: Thank you for your question. We checked all the membranes on MFN1 and MFN2, and found only one of them showed returned by 24 h. And also according to Fig 5B, MFN2 decreased time-dependently. Thus, we replaced membranes of MFN1 and MFN2 in Fig 1B.

  1. Q: Fig 4B is supposed to be a representative set of data presented in Fig 4C. But according to 4C the LC3-II/LC3-I ratio is approximately 1 for the 0 point which is not supported by 4B.

A: Thank you for your question. Actually it is a normalized ratio in 4C. The 0 point is considered as a control group which is normalized as 1.

  1. Q: Many of the methodological descriptions are not sufficient for reproduction of the experiments. E.g. the cell number used for the assay is not given.

A: We agree with you and have incorporated this suggestion throughout our paper.

  1. Q: Lines 384-386: “A Student's two-tailed t-test was used to compare two different groups …” but there is no presented result in which 2 conditions are compared.

A: We agree with your assessment. After double-check, there’s no presented result in which 2 conditions are compared. So, we deleted statistic methods about “Student's two-tailed t-test”. (Line 396-397)

  1. Q: Line 94: “N7 treatment”. What is N7? It is probably a mistake.

A: Thank you for your suggestion and it was a mistake. In a previous paper by our group, NJXA was named as N7(Biochem Pharmacol, 2016, 100: 61-72.). The mistake is already corrected. (Line 94)

  1. Q: The quality of Fig 4E is quite poor. An improved version is expected with higher resolution.

A: Thank you for your suggestion and We’ve tried our best to improve the quality of the picture

  1. Q: Line 166: definition of OC is missing. Probably it refers to the tested compound.

A: Thank you for your suggestion and it was another mistake. The methods of colocalization is refer to another article by our group(Autophagy, 2014, 10(5): 736-749.). The mistake is already corrected. (Line 172)

  1. Q: Some of the cited papers are inefficiently given in the References. In some cases, the journal is missing (e.g. refs 13, 19, 20, 27). A complete and careful checking is suggested.

A: We agree with you and have incorporated this suggestion throughout our paper.

Reviewer 2 Report

The paper is well written and presents a great content about a natural product that promotes mitophagy in cervical cancer cells. I would recommend to publication after some minor consideration, as below:

Page 2 Line 94: what does “n7” mean (seen also in Figure 4D)? I believe this was a code for NJXA.

For me, there are two sentences that do not corroborate each other. At page 9, Line 275 it is written: “The effect of NJXA on inducing mitophagy and inhibiting cancer cell proliferation was abolished by knocking-down the mitophagy regulator ATG7”. At the same page 9, line 280 we see: “Notably, our results also showed that blocking mitophagy by knocking-down the autophagic protein ATG7, the effect of NJXA on inhibiting cell proliferation could not be completely eliminated”. Could the authors please clarify it better?

Page 10 line 306: “according to our results” is too close to the sentence at line 307: “according to our research”. Please, reformulate this sentence.

Page 10, line 310. “These results implied that NJXA selectively targets these cells under starvation conditions and might not be toxic to CERVICAL CANCER CELLS cells under normal nutrient environments at low concentrations”. I suggest to insert “cervical cancer cells” at this point of the text to emphasize that this is about cancer cells, not a speculation about NJXA toxicity in “non-tumoral” cells.

Page 10 Line 317: please detail the process of Nujiangexanthone A purification from Garcinia nujiangensis leaves and provide chromatograms or NMR spectra that evidences the purity. If it has been already published, please provide the reference at the text.

Author Response

Reviewer 2

  1. Q: Page 2 Line 94: what does “n7” mean (seen also in Figure 4D)? I believe this was a code for NJXA.

A: Thank you for your suggestion and it was a mistake. In a previous paper by our group, NJXA was named as N7(Biochem Pharmacol, 2016, 100: 61-72.). The mistake is already corrected. (Line 94)

  1. Q: For me, there are two sentences that do not corroborate each other. At page 9, Line 275 it is written: “The effect of NJXA on inducing mitophagy and inhibiting cancer cell proliferation was abolished by knocking-down the mitophagy regulator ATG7”. At the same page 9, line 280 we see: “Notably, our results also showed that blocking mitophagy by knocking-down the autophagic protein ATG7, the effect of NJXA on inhibiting cell proliferation could not be completely eliminated”. Could the authors please clarify it better?

A: Thanks for the question you raised. The word “abolish” was too absolute for the reversion effect of ATG7-knocking-down on NJXA’s effect. Thus, we changed “abolished” into “attenuated” for better understanding. (Line 216)

  1. Q: Page 10 line 306: “according to our results” is too close to the sentence at line 307: “according to our research”. Please, reformulate this sentence. Already changed the second “according to our research” into on the other hand

A: We agree with you and changed the description. (Line 316-321)

  1. Q: Page 10, line 310. “These results implied that NJXA selectively targets these cells under starvation conditions and might not be toxic to CERVICAL CANCER CELLS cells under normal nutrient environments at low concentrations”. I suggest to insert “cervical cancer cells” at this point of the text to emphasize that this is about cancer cells, not a speculation about NJXA toxicity in “non-tumoral” cells. (Line 321)

A: Thanks for your suggestion and we’ve already added the words as you suggested.

  1. Q: Page 10 Line 317: please detail the process of Nujiangexanthone A purification from Garcinia nujiangensis leaves and provide chromatograms or NMR spectra that evidences the purity. If it has been already published, please provide the reference at the text.

A: Thank you for your suggestion. We’ve already added the reference to the material information about NJXA on line 329.

Reviewer 3 Report

According to the authors, cells under condition of nutrient starvation were more susceptible to xanthone than those cells cultured in a complete medium. The above could raise two questions. One, given that cells cultured in complete medium, it would be expected that they were not under nutrient deprivation conditions, then what kind of stress would these cells have to induce mitophagy? Second, would it be possible to add xanthone in high doses to cells cultured in complete medium induced mitophagy?

MitoTracker, a fluorescent dye that directly accumulates on active mitochondria, was used to detect the mass of mitochondria. NJXA markedly reduced the red fluorescence intensity of the dye, indicating decreased mitochondrial mass. Even more than 20 μM CCCP, a mitophagy inducer, the effect of NJXA on decreasing fluorescence was more significant. However, in fig 2C, it is also observed that the activity of DMSO is almost the same as CCCP. The above would indicate that DMSO is active and should be considered since it was the solvent used. The other possibility is that in this particular case, the CCCP was not as active as expected. The authors could explain these possible inconsistencies.

On the other hand, NJXA has 4 alcohols of the phenol type, which would mean a ROS scavenger. Likewise, it would indicate that this xanthone could be acid. However, the most important thing is that the pH of the different experiments did not induce the ionization of this xanthone since due to metabolically active mitochondria, they have highly negative membrane potential. Charged molecules are generally unable to cross cell membranes without the aid of transporter proteins due to the considerable activation energies associated with removing associated water molecules. This effect could be present in cells under complete media where the outer membrane is functionally active and not in cells under nutrient starvation where possibly the outer membrane is damage.

Considering the above, would it be appropriate for the authors to indicate the pH conditions in the different experiments, and if it were pH7, would this ensure that the xanthone is not ionized?

Author Response

Reviewer 3

  1. Q: According to the authors, cells under condition of nutrient starvation were more susceptible to xanthone than those cells cultured in a complete medium. The above could raise two questions. One, given that cells cultured in complete medium, it would be expected that they were not under nutrient deprivation conditions, then what kind of stress would these cells have to induce mitophagy? Second, would it be possible to add xanthone in high doses to cells cultured in complete medium induced mitophagy?

A: Thank you for your question. According to our results, NJXA caused reduction in mitochondrial membrane potential is the stress that leading to mitophagy in complete medium. And our study showed that NJXA at 20uM for 4h can effectively induced mitophagy in complete medium. For nutrient starvation condition, the effect is just more sensitive.

  1. Q: MitoTracker, a fluorescent dye that directly accumulates on active mitochondria, was used to detect the mass of mitochondria. NJXA markedly reduced the red fluorescence intensity of the dye, indicating decreased mitochondrial mass. Even more than 20 μM CCCP, a mitophagy inducer, the effect of NJXA on decreasing fluorescence was more significant. However, in fig 2C, it is also observed that the activity of DMSO is almost the same as CCCP. The above would indicate that DMSO is active and should be considered since it was the solvent used. The other possibility is that in this particular case, the CCCP was not as active as expected. The authors could explain these possible inconsistencies.

A: Thank you for your suggestion and we’ve already added the explanation as you suggested. (Line 114-118)

  1. Q: On the other hand, NJXA has 4 alcohols of the phenol type, which would mean a ROS Likewise, it would indicate that this xanthone could be acid. However, the most important thing is that the pH of the different experiments did not induce the ionization of this xanthone since due to metabolically active mitochondria, they have highly negative membrane potential. Charged molecules are generally unable to cross cell membranes without the aid of transporter proteins due to the considerable activation energies associated with removing associated water molecules. This effect could be present in cells under complete media where the outer membrane is functionally active and not in cells under nutrient starvation where possibly the outer membrane is damage. Considering the above, would it be appropriate for the authors to indicate the pH conditions in the different experiments, and if it were pH7, would this ensure that the xanthone is not ionized?

A: Thank you for your suggestion. We have already measured the pH of the main conditions in our experiments. NJXA treatment did not significantly changed the pH in both complete medium or serum withdraw condition. (CM-DMSO: 8.28±0.02; CM-NJXA: 8.21±0.04; SW-DMSO: 8.13±0.03; SW-NJXA: 8.12±0.03; HeLa cells were treated with 20 μM of NJXA or DMSO for 4 h). On the other hand, the dosage of NJXA we used in those experiments were very low (1:1000 diluted at most), so the acid of compound might not be able to influence the pH of medium. From the data above, NJXA does not seem to function through active transport into the cells. Another possibility is that it binds to receptors on the cell membrane to regulate a series of signaling pathways. However, the above hypothesis cannot be clarified by the current research, and the pharmacokinetics of NJXA still needs more in-depth follow-up research.

Round 2

Reviewer 1 Report

Opinion 

on the revised version of the manuscript by Ji-ling Feng et al., entitled “Nujiangexanthone A inhibits cervical cancer cell proliferation by promoting mitophagy”

Manuscript ID: molecules-1179931

The manuscript has been substantially improved, Authors corrected all the points objected before. Based on these improvements the acceptance of the revised version is suggested.